# Preoperative Biomarkers and Survival in Chinese Breast Cancer Patients with HIV: A Propensity-Score-Matched-Cohort Study

**DOI:** 10.3390/v15071490

**Published:** 2023-06-30

**Authors:** Qian Wu, Li Deng, Ye Cao, Shixian Lian

**Affiliations:** 1Department of General Surgery, Fudan University Affiliated Huadong Hospital, Shanghai 200040, China; 17211210092@fudan.edu.cn; 2Department of General Surgery, Shanghai Public Health Clinical Center, Fudan University, Shanghai 201508, China; dengli@shphc.org.cn (L.D.); caoye@shphc.org.cn (Y.C.)

**Keywords:** breast neoplasms, HIV, prognosis, biomarkers, CD3

## Abstract

Background: China initiated its national free antiretroviral therapy program in 2004 and saw a dramatic decline in mortality among the population with HIV. However, the morbidity of non-AIDS-defining cancers such as breast cancer is steadily growing as life expectancy improves. The aim of this study was to investigate the clinical characteristics and prognosis of breast cancer patients with HIV in China. Materials and methods: Data from 21 breast cancer patients with HIV and 396 breast cancer patients without HIV treated at the Shanghai public health clinical center from 2014–2022 was collected. After propensity score matching, 21 paired patients in the two groups were obtained and compared. The optimal cut-off value of preoperative biomarkers for recurrence was determined via maximally selected log-rank statistics. Preoperative biomarkers were categorized into high and low groups, based on the best cut-off values and compared using Kaplan–Meier survival curves and the log-rank test. The Cox proportional hazards regression model was used to perform univariate and multivariate analyses. Results: The median follow-up time was 38 months (IQR: 20–68 months) for the propensity-score-matching cohort. The progression-free survival at 1, 2 and 3 years for patients with and without HIV were 74.51%, 67.74%, and 37.63% and 95.24%, 95.24%, and 90.48%, respectively. The overall survival for patients with HIV at 1, 2 and 3 years were 94.44%, 76.74%, and 42.63%. After multivariate analysis, Only HIV status (hazard ratios (HRs) = 6.83, 95% [confidence intervals (CI)] 1.22–38.12) were associated with progression-free survival. Based on the best cut-off value, CD8 showed discriminative value for overall survival (*p* = 0.04), whereas four variables, the lymphocyte-to-monocyte ratio (*p* = 0.02), platelet-to-lymphocyte ratio (*p* = 0.03), CD3 (*p* = 0.01) and CD8 (*p* < 0.01) were suggested be significant for progression-free survival. The univariate analysis suggested that CD3 (HRs = 0.10, 95% [CI] 0.01–0.90) and lymphocyte-to-monocyte ratio (HRs = 0.22, 95% [CI] 0.05–0.93) were identified as significant predictors for progression-free survival. Conclusion: In this study, breast cancer in patients with HIV in China reflected a more aggressive nature with a more advanced diagnostic stage and worse prognosis. Moreover, preoperative immune and inflammatory biomarkers might play a role in the prognosis of breast cancer patients with HIV.

## 1. Introduction

Ever since the introduction and widespread application of the potent combination antiretroviral therapy (cART) in 1996, the fatality ratio for AIDS-defining cancers has considerably diminished, to a substantial extent [1]. However, as people living with HIV continue to enjoy an extended lifespan, the morbidity rate of non-AIDS-defining cancers has gradually risen [2], with the tumor spectrum increasingly aligning with that of the general population [3]. In recent years, breast cancer has emerged as the predominant malignancy among women [4], with the United States witnessing a ten-fold rise in the absolute number of breast cancer cases among people living with HIV between 1991and 1995 and 2001and 2005 [5]. China initiated its nationwide ART program as recently as 2004, leading to a striking decline in HIV mortality rates over the succeeding decades [6]. Despite numerous studies [7,8,9] examining the epidemiology and prognosis of breast cancer among people living with HIV in Africa and North America, there is still a woeful lack of information concerning the situation in China.

The immune system of people with HIV is suppressed before the administration of cART. Even after the medication is prescribed, the lingering chronic immune alteration has the potential to impact the local immune response, according to recent research [10]. In the field of tumor biology, distinct types of immune cells may contribute differently. CD3 is present on almost all T cells, and a higher CD3 expression is believed to indicate better prognosis in breast cancer [11]. Furthermore, CD4 lymphocytes (the primary targets of HIV) are indicative of immune function, with their quantities reflecting immune function levels to a significant extent. Generally, higher CD4 counts are associated with stronger immune function [12]. In fact, prior studies [13,14] have evidenced a high correlation between degree of immunosuppression based on CD4 cell counts and the incidence and prognosis of AIDS-defining cancers. Additionally, CD8 lymphocytes with their cytotoxic effects and cytokine excretion exhibit anti-tumor behavior, which was demonstrated in colorectal, esophageal, and pancreatic cancers [15,16,17]. The CD4/CD8 ratio is a monitoring indicator for immune recovery in patients with HIV undergoing treatment. It is widely recognized to be positively associated with improved prognosis [18]. However, the connection between breast cancer and immunosuppression in patients with HIV remains largely unexplored.

In addition to immunodeficiency, patients with HIV experience chronic inflammation and immune activation, despite effective viral suppression. The tumor microenvironment comprises various cellular components, including monocytes, neutrophils, platelets, and lymphocytes, which collectively influence tumor growth and development. Specifically, neutrophils and monocytes have been observed to promote tumor progression and facilitate primary tumor evasion [19,20]. Conversely, lymphocytes exhibit anti-tumor effects [21]. Platelets, on the other hand, often represent a non-specific response to inflammation, which can contribute to the promotion of tumor cell growth [22]. Biomarkers reflecting systemic inflammation, such as the neutrophil-to-lymphocyte ratio, lymphocyte-to-monocyte ratio, and platelet-to-lymphocyte ratio, have been extensively studied and are widely recognized for their association with tumor prognosis [23,24,25,26]. Furthermore, the efficacy of inflammatory biomarkers in predicting the prognosis of non-AIDS-defining cancers and AIDS-defining cancers among patients with HIV has also been certified [27]. However, there is a dearth of research investigating the correlation between biomarkers such as the lymphocyte-to-monocyte ratio and the prognosis of breast cancer in patients with HIV. It is currently unclear if the inflammatory response of tumors among individuals with chronic immunosuppression is comparable to that of the general population.

Thus, the primary objective of this study was to examine the clinical characteristics and subsequent prognosis of breast cancer in Asian persons living with HIV, as opposed to that of the general population. Moreover, our endeavor was to delve deeply into the relationship between immune and inflammatory biomarkers in the peripheral blood and the overall prognosis in the aforementioned populace.

## 2. Materials and Methods

### 2.1. Patients

The clinical data of patients who were pathologically diagnosed with breast cancer between July 2014 and January 2022 at the Shanghai public health clinical center were collected consecutively and analyzed retrospectively. A total of 417 patients were included, which were divided into the patients-with-HIV group (*n* = 21) and patients-without-HIV group (*n* = 396). The inclusion criteria were: 1. Confirmed pathological diagnosis of breast cancer; 2. HIV infection status determined through laboratory tests, including enzyme-linked immunosorbent assay (ELISA) and Western blot analysis; 3. Mandatory HIV assessment before commencing treatment for breast cancer; 4. Comprehensive collection of clinical characteristics and follow-up data (including patients’ age, race, gender, menopausal status, tumor stage, molecular type, and preoperative blood test (neutrophils, lymphocyte, monocyte, platelet, CD4 cell counts, CD8 cell counts, and CD3 cell counts) from HIV patients, as immunological indicators were not routinely assessed preoperatively in patients without HIV. All patients were females. The ethical approval of this study was granted by the ethics committee of the Shanghai public health clinical center (No: 2022-S087-02). The Shanghai public health clinical center ethics committee waived the need for informed consent from patients, since it was a retrospective study.

### 2.2. Follow-Up

Postoperative follow-up was performed every 3 months for 2 years after surgery and every 6 months thereafter, according to the National Comprehensive Cancer Network guidelines. The criteria for diagnosing recurrence or metastasis were: 1. Pathological diagnosis using puncture biopsy; 2. For patients who failed to undergo a biopsy of suspected metastases, the definitive diagnosis of recurrence was established by imaging modalities such as computer tomography, magnetic resonance imaging, mammography and ultrasound or blood tests such as tumor markers. The progression-free survival time was defined as the interval between cancer diagnosis and time of recurrence, disease progression, or death from any cause. The overall survival time was defined as the interval between cancer diagnosis and time of breast cancer-related death.

### 2.3. Statistical Analysis

Categorical variables were reported as integers and proportions, and continuous variables were reported as median (interquartile range (IQR)) or mean (standard deviation (SD)), as appropriate. Propensity score matching was utilized to achieve a covariate balance between the two groups and reduce confounding factors. The propensity score was defined here as the probability of living with HIV versus living without HIV in breast cancer patients with clinicopathological characteristics. It was estimated using the logistic regression model that had been established from the factors potentially affecting the prognosis, including age at diagnosis, stage and molecular subtype. The propensity score matching was performed using 1:1 nearest-neighbor matching without setting the maximum caliper, due to the small sample of patients from the HIV group [28]. The optimal cut-off value of preoperative biomarkers for recurrence was determined via maximally selected log-rank statistics [29]. Preoperative biomarkers were categorized into high and low groups, based on the best cut-off values and compared using Kaplan–Meier survival curves and the log-rank test. The association of the clinicopathologic characteristics between the two groups was analyzed using Fisher’s exact test or the Mann–Whitney U test, as appropriate. The Cox proportional hazards regression model was used to perform univariate and multivariate analyses, which provided estimates of hazard ratios (HRs), their 95% confidence intervals (95% CIs), and *p* values using the Wald test. All statistical analyses were determined using the R software (version 4.1.2, http://www.r-project.org, accessed on 1 November 2021). The R packages “gmodels”, “survival”, “survminer” and “matchit” were used. A two-sided *p* < 0.05 was considered statistically significant.

## 3. Results

### 3.1. Demographic and Clinicopathological Characteristics

The study included a cohort of 417 breast cancer patients, who were stratified into two groups based on their HIV status: a cohort of 21 patients with HIV and a cohort of 396 people without HIV. The clinicopathological characteristics of the two groups before propensity score matching were outlined in Table 1, revealing significant differences between patients with and without HIV. Specifically, patients with HIV had larger tumor diameter (*p* < 0.01), exhibited higher HER-2 positivity (*p* = 0.04), and presented with a more advanced tumor stage (*p* = 0.03) compared to those without HIV. However, there were no significant discrepancies in other factors, such as age, menstrual status, hormone receptor status, lymph node involvement, and molecular subtype between the two groups. A total of 21 patients with HIV and 21 patients without HIV were included after propensity score matching (Table 2). A statistically significant difference in tumor size remained between the two groups, even after propensity score matching (*p* = 0.01). However, upon dividing the tumor size into three categories based on size and conducting a Fisher’s exact test, no statistically significant difference was found between the two groups (*p* = 0.55). This might be attributed to the larger size of tumors in patients with HIV at the onset of diagnosis and the limited sample size of this group. In contrast, there were no significant variations in other baseline characteristics across groups.

Out of the 21 patients diagnosed with HIV, the median duration between their HIV diagnosis and breast cancer diagnosis was 19 months (IQR: 1–66 months). Due to limited availability of initial information at the time of HIV infection, blood test data was collected during the patient’s diagnosis of breast cancer. Immunological markers, such as CD3, were not routinely tested in patients without HIV before their surgery; therefore, data from those without HIV was not included in the study. The mean CD4 was 369.33 ± 225.44/μL and the mean CD4/CD8 was 0.78 ± 0.77. The mean neutrophil-to-lymphocyte ratio, lymphocyte-to-monocyte ratio and platelet-to-lymphocyte ratio was 2.39 ± 1.91, 4.58 ± 2.93 and 154.09 ± 70.60, respectively. All patients were treated with standardized cART. The clinical immune and inflammatory biomarker features of patients with HIV are summarized in Table 3.

### 3.2. Prognosis Comparison between Patients with and without HIV

The median follow-up time was 38 months (IQR: 20–68 months) for the propensity-score-matching cohort (*n* = 42). At the end of follow-up, five patients died and three patients progressed in the group with HIV, compared with one death and two progressions in the group without HIV. Patients with HIV demonstrated progression-free survival rates of 74.51%, 67.74%, and 37.63% at the end of 1, 2, and 3 years, respectively, whereas patients without HIV demonstrated relatively higher progression-free survival rates of 95.24%, 95.24%, and 90.48%, for the same time intervals. The log-rank test affirmed that HIV infection represented a statistically significant risk factor for disease progression in breast cancer (*p* < 0.01). Additionally, the overall survival rates for patients with HIV were 94.44%, 76.74%, and 42.63% at 1, 2, and 3 years, compared with 100%, 100% and 100% for patients without HIV during the same time period. And the six-year survival rate for patients without HIV was as high as 90.00%. The log-rank test demonstrated that patients with HIV had a poorer prognosis than patients without HIV (*p* < 0.01). Figure 1 illustrates the survival curves for overall survival and progression-free survival.

Univariate and multivariate COX regression analyses were then conducted on the propensity score matching data (Table 4). Hormone receptor status (hazard ratios (HRs) = 0.15, 95% [confidence intervals (CI)] 0.03–0.71), tumor size (HRs = 3.36, 95% [CI] 1.26–8.97), lymph node involvement (HRs = 10.29, 95% [CI] 1.31–80.73), molecular subtype (HRs = 1.99, 95% [CI] 1.01–3.92), tumor stage (HRs = 4.73, 95% [CI] 1.38–16.19) and HIV status (HRs = 8.26, 95% [CI] 1.64–41.66) were found in association with breast cancer disease progression by a univariate COX analysis. Only HIV status (HRs = 6.83, 95% [CI] 1.22–38.12) was associated with progression-free survival after the multivariate analysis. Notably, COX regression analysis did not uncover any statistically significant association between any of the variables and overall survival.

### 3.3. Prognosis Role of Preoperative Biomarkers in Breast Cancer Patients with HIV

The optimal cut-off values of the preoperative biomarkers and the log-rank values calculated from maximally selected log rank statistics for the breast cancer patients with HIV are shown in Appendix A, and the distribution plot of overall survival and progression-free survival are shown in Appendix A. The maximally selected log-rank analysis revealed that CD8 cell counts (*p* = 0.04) may potentially be related to overall survival, while lymphocyte-to-monocyte ratio (*p* = 0.02), platelet-to-lymphocyte ratio (*p* = 0.03), CD3 cell counts (*p* = 0.01), and CD8 cell counts (*p* < 0.01) could be potentially linked with progression-free survival in breast cancer patients affected by HIV. Kaplan–Meier survival curves were used to compare high and low groups, separated based on the optimal cut-off values of preoperative biomarkers for overall survival and progression-free survival. The lower CD8 group (<758.00) exhibited a relatively poor overall survival compared to the higher group (Figure 2). In terms of progression-free survival, patients with higher CD3 (>1033.80), higher CD8 (>662.70), higher lymphocyte-to-monocyte ratio (>3.59) and lower platelet-to-lymphocyte ratio (98.17) had a relatively better prognosis (Figure 3).

The associations between clinical immune and inflammatory biomarkers and the prognosis of breast cancer patients affected by HIV through COX regression analysis are demonstrated in Table 5. In the univariate analysis, the lymphocyte-to-monocyte ratio (HRs = 0.22, 95% [CI] 0.05–0.93) and the CD3 cell count (HRs = 0.10, 95% [CI] 0.01–0.90) were considered to be associated with progression-free survival outcomes. However, multivariate analysis failed to indicate any statistically significant association between these variables and progression-free survival outcomes. Additionally, no statistically significant association between these variables and overall survival outcomes was recorded in the univariate analysis.

## 4. Discussion

In this study, our primary focus was to conduct an exploratory clinicopathological and prognostic analysis of the breast-cancer-patient cohort living with and without HIV in a solitary Asian center. Our findings indicated that patients with concomitant HIV infection exhibited a more advanced stage of cancer at diagnosis and a substantially worse prognosis. Multivariate analyses revealed that only HIV infection demonstrated a significant correlation with a breast cancer prognosis. Additionally, our study’s preliminary investigation of pre-treatment biomarkers has identified several biomarkers that warrant further examination in subsequent studies, to assess their potential prognostic value. A lowered platelet-to-lymphocyte ratio, an increased lymphocyte-to-monocyte ratio, an elevated CD8 count, and a higher CD3 count might all contribute positively to a better prognosis.

Previous research has shown that people living with HIV were diagnosed with breast cancer at a relatively younger age compared to those without HIV [9,10,30]. Furthermore, a younger age has been shown to be associated with a more aggressive disease upon diagnosis [31]. Consequently, people living with HIV tend to present with biologically more aggressive breast cancer, resulting in later stages of diagnosis and worse survival outcomes [10]. Our study showed that patients with HIV and breast cancer had an average age of 51.05, approximately 47.62% of whom were under the age of 50. No noteworthy difference was observed between the age of the two groups, conceivably due to the limited data available on HIV patients. Among the total number of patients diagnosed, seven (33.33%) were found to be in advanced stages (III and IV). Likewise, a study by Cubasch [9] revealed that over half of all HIV-positive breast cancer patients were diagnosed at an advanced stage, with around 78.00% being under the age of 50. Similarly, Coghill [32] analyzed 1197 breast cancer patients with HIV in the United States, finding that stage III and IV breast cancer accounted for nearly 37.20% of cases.

In recent years, the 5-year overall survival rate of breast cancer in the Chinese population has reached more than 80.00% [33]. In our study, while we acknowledge the potential bias due to the small sample size, it is worth noting that the 6-year overall survival rate of breast cancer patients without HIV matched by propensity score matching reached 90.00%, which provides informative insights into the long-term outcomes of this cohort. In contrast, the overall survival and progression-free survival rates at three years for patients diagnosed with HIV in our study registered as low as 42.63% and 37.63%, respectively. Univariate and multivariate analyses demonstrated that HIV infection was a significant prognostic factor associated with poorer outcomes. Limited access to general mammography screening, a higher likelihood of not seeking timely treatment when issues are detected, and poorer adherence to treatment among patients with HIV might collectively contribute to the less-favorable prognosis observed in this population. While our study indicated a potential association between HIV and a poorer prognosis in breast cancer patients, it is crucial to emphasize that larger-scale data are necessary to establish a definitive conclusion.

Inflammation-based biomarkers, such as the lymphocyte-to-monocyte ratio, have been confirmed to be associated with the prognosis of various tumors, including colorectal cancer [24], gastrointestinal tumors [25], and renal carcinomas [26]. It is well-established that the induction of tumorigenesis can trigger a systemic inflammatory response characterized by an upregulation of cell proliferation, the production of toxic reactive oxygen species, and a series of other deleterious consequences [34]. The influence of different inflammatory cells on the tumor environment can yield opposing effects. On one hand, monocytes and macrophages promote the proliferation and advancement of tumors, thereby promoting the migration of breast cancer cells, creating a vascular endothelial network, and consequently creating a conducive environment for tumor survival [19]. On the other hand, lymphocytes have been observed to play a crucial role in tumor monitoring and editing, and exhibit an anti-tumor effect [21]. Our study found that preoperative inflammation-based biomarkers, such as lymphocyte-to-monocyte ratio, exhibited predictive potential for the prognosis of breast cancer patients with HIV. Similar to in the general population, a higher lymphocyte-to-monocyte ratio value was indicative of better clinical outcomes. The elevated lymphocyte-to-monocyte ratio score reflected a higher concentration of lymphocytes and a lower level of monocytes in the peripheral blood, leading to an unfavorable environment for tumor growth.

Additionally, our findings indicate that the cut-off thresholds of biomarkers appear to be lower in patients with HIV when compared to the general population. Yin [35] posited that individuals with a lymphocyte-to-monocyte ratio >4.85 in the general population tend to have a more favorable prognosis, while our study suggests a cut-off value of 3.59 for lymphocyte-to-monocyte ratio. Regrettably, a statistical comparison could not be performed, due to the unavailability of preoperative peripheral blood markers from patients without HIV. Studies conducted previously have indicated that chronic HIV infection can result in permanent alterations in the lymph node structure due to perturbations in the CD4 homeostasis, even with cART administration [36]. In situations of prolonged immune suppression, reduced lymphocyte-to-monocyte ratio values can still indicate a certain level of anti-tumor defense. Comparable lymphocyte-to-monocyte ratio cut-off values have been identified in previous studies carried out on patients with HIV. For instance, Zeng et al. [37] highlighted a strong association between a diminished lymphocyte-to-monocyte ratio (<2.74) and unsatisfactory survival outcomes in individuals with diffuse large B-cell lymphoma. Of course, a sophisticated randomized controlled trial would be necessary to lend weight to these results.

CD8 lymphocytes are widely recognized for their significant contribution in reducing HIV replication and enhancing immunity [38]. Prior immunohistochemical studies [39,40] have demonstrated the ability of CD8 lymphocytes to interact with tumor cells, resulting in the production of interferons, which in turn triggered a series of antitumor responses, including cell cycle inhibition and the induction of macrophage tumoricidal activity. As a result, several studies [40,41] have suggested that a higher infiltration of CD8 is associated with a more favorable prognosis among breast tumor patients. Our study also observed a positive correlation between CD8 and the prognosis of breast cancer in patients with HIV using the Kaplan–Meier approach. However, no significant association was observed between them in a univariate analysis. It is important to note that this paradox of association may be attributed to the limitations posed by our small sample size. Consequently, it becomes imperative to expand the sample size in future studies to conduct a more comprehensive assessment of this association.

CD3 is expressed in all developmental stages of T lymphocytes, and is present in various subpopulations, including Treg cells, CTL cells, TH1 cells and TH2 cells, each exhibiting different interactions with the tumor targets. TH and Treg cells secrete cytokines such as IL-2 that regulate the immune response and inhibit tumors [42]. People living with HIV experience severe loss of CD3 T cell homeostasis [43]; however, previous studies found that after long-term use of cART drugs, the percentage of CD3 T cells remained relatively constant, despite significant alterations in the CD4 and CD8 T cell subsets [44]. Previous studies have investigated CD3 in the context of tumor prognosis. Galon et al. [15]. suggested that CD3 displayed a stronger prognostic value as compared to CD8 in colon cancer. Similarly, Savas et al. [45]. considered that higher CD3 T cell numbers indicated a more robust antigen-experienced, anti-tumor immune response, and were associated with better breast cancer prognosis. Our study indicates that CD3 may have a predictive role for breast cancer patients with HIV. Our univariate analysis revealed a positive correlation between CD3 and breast cancer prognosis. However, the association did not remain significant in the multivariable analysis. It is imperative to conduct additional experiments to reconfirm our conclusions, particularly in patients with HIV undergoing specific mechanistic changes. Furthermore, it is essential to acknowledge that the data cannot be presented as statistically significant. The limited sample size likely influenced these outcomes, indicating that the study was underpowered to detect a significant difference.

This study boasts several strengths compared to prior research on breast cancer patients in the Chinese HIV population. To date, there have been few investigations into this area, which is hindered by insufficient knowledge and the discriminatory attitudes towards HIV that persist in China. Countless individuals tend to reject treatment or show poor compliance upon discovering their illness. Consequently, it is imperative to not only educate the populace about HIV, but also to conduct regular breast cancer screenings among women >40 with HIV, as recommended by the European AIDS Clinical Society [46]. Improvements are necessary to ensure that women living with HIV are diagnosed with breast cancer at an earlier stage, so that they can receive timely and appropriate care for a breast cancer diagnosis. Moreover, our study delved into the prognostic value of preoperative immune and inflammatory biomarkers on breast cancer, a promising area of exploration that could only provide preliminary findings yet succeeded in drawing renewed attention from scholars via this article.

Several limitations were also identified in this article. Firstly, the study utilized a retrospective design. Secondly, the number of patients included was relatively small, and the follow-up duration was comparably brief. This limited sample size gave rise to the possibility of bias, thereby impacting the robustness and reliability of the results. Lastly, despite the implementation of a propensity-score-matching approach to equalize the clinical characteristics of the two groups, a significant difference in tumor size persisted, which could potentially undermine the conclusions drawn. For greater confidence in the legitimacy of the results, future studies should encompass a more expansive study population, an extended follow-up period, and a multicenter trial setting.

## 5. Conclusions

In conclusion, our exploratory study elucidated the fact that breast cancer in the Chinese HIV population manifests a more aggressive nature, with advanced diagnostic stages and poorer prognoses. Furthermore, preoperative immune and inflammatory biomarkers may influence the prognosis of breast cancer patients with HIV and need to be further investigated. The correlation between HIV infection status and breast cancer prognosis, the interplay between cART and postoperative adjuvant therapy, and the optimal standards of care for this specific demographic merit clinical consideration. We urgently recommend intensifying breast cancer screening in women with HIV to amplify the prognosis of breast cancer patients with HIV. We fervently hope to stimulate subsequent research to explore broader horizons in this realm and to improve outcomes for breast cancer patients with HIV.

## Figures and Tables

**Figure 1 viruses-15-01490-f001:**
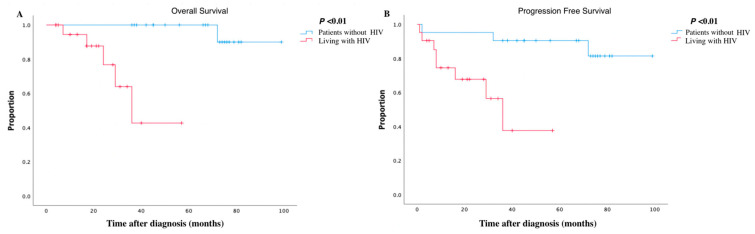
Kaplan–Meier analyses in breast cancer patients with HIV and without HIV. (**A**) showing overall survival of breast cancer; (**B**) showing progression-free survival of breast cancer.

**Figure 2 viruses-15-01490-f002:**
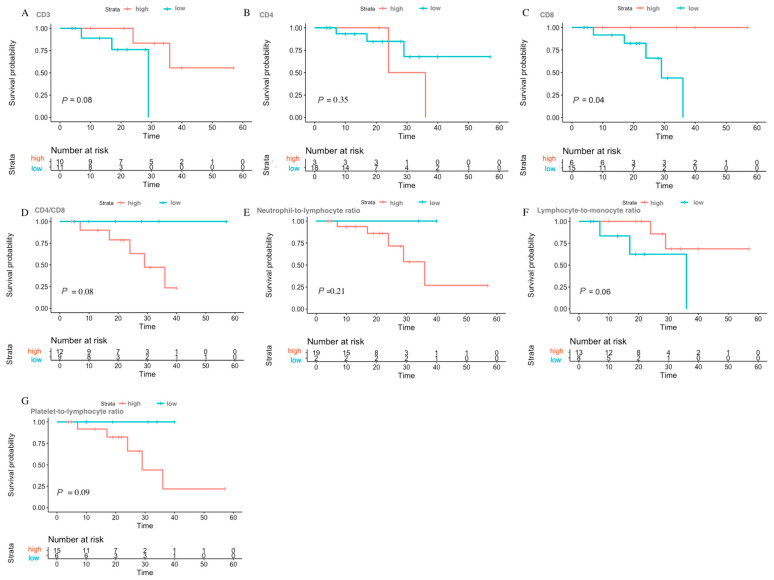
Prognosis value of different preoperative biomarkers for overall survival in breast cancer patients with HIV. (**A**) CD3; (**B**) CD4; (**C**) CD8; (**D**) CD4/CD8 ratio; (**E**) neutrophil-to-lymphocyte ratio; (**F**) lymphocyte-to-monocyte ratio; (**G**) platelet-to-lymphocyte ratio.

**Figure 3 viruses-15-01490-f003:**
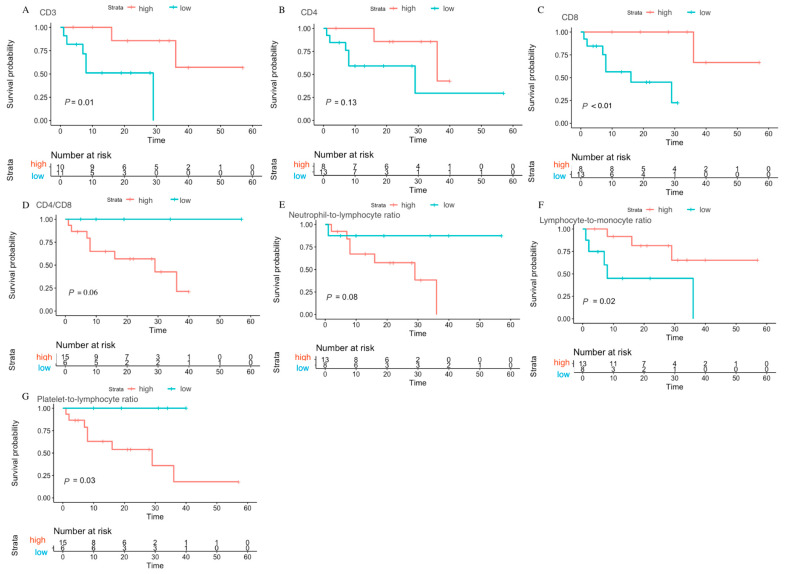
Prognosis value of different preoperative biomarkers for progression-free survival in breast cancer patients with HIV. (**A**) CD3; (**B**) CD4; (**C**) CD8; (**D**) CD4/CD8 ratio; (**E**) neutrophil-to-lymphocyte ratio; (**F**) lymphocyte-to-monocyte ratio; (**G**) platelet-to-lymphocyte ratio.

**Table 1 viruses-15-01490-t001:** Clinical characteristics before propensity score matching (*n* = 417).

Clinical Characteristics		HIV (*n* = 21)	Non-HIV (*n* = 396)	*p* Value
Age	Mean (SD)	51.05 (10.40)	55.14 (12.22)	0.13
	<50	10 (47.62%)	134 (33.84%)	
	≥50	11 (52.38%)	262 (66.16%)	0.20
Menopause	no	9 (42.86%)	131 (33.08%)	
	yes	12 (57.14%)	265 (66.92%)	0.36
Hormone receptor	positive	12 (57.14%)	268 (67.68%)	
	negative	9 (42.86%)	128 (32.32%)	0.32
Her-2	positive	9 (42.86%)	91 (22.98%)	
	negative	12 (57.14%)	305 (77.02%)	0.04
Tumor size (cm)	Mean (SD)	4.40 (2.47)	2.51 (1.29)	<0.01
	t > 5 cm	3 (14.29%)	12 (3.03%)	
	5 ≥ t > 2 cm	13 (61.90%)	195 (49.24%)	
	t ≤ 2 cm	5 (23.81%)	189 (47.73%)	0.01
Lymph node involvement	no	8 (38.10%)	220 (55.56%)	
	yes	13 (61.90%)	176 (44.44%)	0.12
Molecular type	Luminal A	1 (4.76%)	51 (12.88%)	
	Luminal B	11 (52.38%)	217 (54.80%)	
	HER-2	5 (23.81%)	50 (12.62%)	
	TNBC	4 (19.05%)	78 (19.70%)	0.44
Stage	0	1 (4.76%)	17 (4.29%)	
	I	2 (9.52%)	111 (28.03%)	
	II	11 (52.38%)	170 (42.93%)	
	III	5 (23.81%)	94 (23.74%)	
	IV	2 (9.52%)	4 (1.01%)	0.03

HIV: human immunodeficiency virus; Her-2: human epidermal growth factor receptor-2; TNBC: triple-negative breast cancer.

**Table 2 viruses-15-01490-t002:** Clinical characteristics after propensity score matching (*n* = 42).

Clinical Characteristics		HIV (*n* = 21)	Non-HIV (*n* = 21)	*p* Value
Age	Mean (SD)	51.05 (10.40)	51.43 (10.16)	0.91
	<50	10 (47.62%)	10 (47.62%)	
	≥50	11 (52.38%)	11 (52.38%)	1.00
Menopause	no	9 (42.86%)	10 (47.62%)	
	yes	12 (57.14%)	11 (52.38%)	0.76
Hormone receptor	positive	12 (57.14%)	12 (57.14%)	
	negative	9 (42.86%)	9 (42.86%)	1.00
Her-2	positive	9 (42.86%)	7 (33.33%)	
	negative	12 (57.14%)	14 (66.67%)	0.53
Tumor size (cm)	Mean (SD)	4.40 (2.47)	2.80 (1.35)	0.01
	t > 5 cm	3 (14.29%)	2 (9.52%)	
	5 ≥ t > 2cm	13 (61.90%)	10 (47.62%)	
	t ≤ 2 cm	5 (23.81%)	9 (42.86%)	0.55
Lymph node involvement	no	8 (38.10%)	11 (52.38%)	
	yes	13 (61.90%)	10 (47.62%)	0.35
Molecular type	Luminal A	1 (4.76%)	1 (4.76%)	
	Luminal B	11 (52.38%)	11 (52.38%)	
	HER-2	5 (23.81%)	5 (23.81%)	
	TNBC	4 (19.05%)	4 (19.05%)	1.00
Stage	0	1 (4.76%)	1 (4.76%)	
	I	2 (9.52%)	5 (23.81%)	
	II	11 (52.38%)	9 (42.86%)	
	III	5 (23.81%)	6 (28.57%)	
	IV	2 (9.52%)	0 (0%)	0.56

**Table 3 viruses-15-01490-t003:** Clinical characteristics of the patients with HIV (*n* = 21).

Clinical Characteristics		HIV (*n* = 21)
Time of HIV diagnosis to breast cancer diagnosis (months)	median (IQR)	19 (1–66)
	≤12	10 (47.62%)
	>12	11 (52.38%)
CD4 counts (/μL)	mean (SD)	369.33 (225.44)
CD4/CD8 ratio	mean (SD)	0.78 (0.77)
CD8 counts (/μL)	mean (SD)	637.48 (297.28)
CD3 counts (/μL)	mean (SD)	1047.10 (366.77)
Neutrophil (10^9^/L)	mean (SD)	2.85 (1.51)
Lymphocyte (10^9^/L)	mean (SD)	1.40 (0.50)
Monocyte (10^9^/L)	mean (SD)	0.37 (0.17)
Platelet (10^9^/L)	mean (SD)	193.48 (67.50)
Neutrophil-to-lymphocyte ratio	mean (SD)	2.39 (1.91)
Lymphocyte-to-monocyte ratio	mean (SD)	4.58 (2.93)
Platelet-to-lymphocyte ratio	mean (SD)	154.09 (70.60)

**Table 4 viruses-15-01490-t004:** Univariate and multivariate survival analyses for propensity-score-matching cohort (*n* = 42).

	Progression-Free Survival				Overall Survival	
	Univariate		Multivariate		Univariate	
Characters	HRs (95% CI)	*p*	HRs (95% CI)	*p*	HRs (95% CI)	*p*
Age (≥50 vs. <50)	0.456 (0.13–1.57)	0.21	-	-	1.62 (0.29–8.97)	0.58
Menopause (yes vs. no)	0.69 (0.21–2.27)	0.54	-	-	4.17 (0.49–35.76)	0.19
Hormone Receptor (+ vs. −)	0.15 (0.03–0.71)	0.02	0.07 (0.01–1.07)	0.06	0.16 (0.02–1.37)	0.09
Her-2 (+ vs. −)	2.26 (0.69–7.45)	0.18	-	-	1.99 (0.40–9.91)	0.40
Tumor size (>5 vs. >2 vs. ≤2)	3.36 (1.26–8.97)	0.02	1.75 (0.54–5.74)	0.35	3.33 (0.85–13.07)	0.08
Lymph node involvement (yes vs. no)	10.29 (1.31–80.73)	0.03	4.18 (0.41–43.03)	0.23	5.07 (0.59–43.70)	0.14
Molecular subtype (TNBC vs Her-2 vs. Hormone Receptor+)	1.99 (1.01–3.92)	0.05	0.46 (0.09–2.43)	0.36	2.07 (0.81– 5.30)	0.13
Stage (III/IV vs. 0/I/II)	4.73 (1.38–16.19)	0.01	1.76 (0.35–8.84)	0.49	4.99 (0.91–27.26)	0.06
HIV (yes vs. no)	8.26 (1.64–41.66)	0.01	6.83 (1.22–38.12)	0.03	322.47 (0.04–2,563,794.24)	0.21

HRs: hazard ratios; CI: confidence intervals.

**Table 5 viruses-15-01490-t005:** Univariate and multivariate survival analyses for breast cancer patients with HIV.

	Progression-Free Survival				Overall Survival	
	Univariate		Multivariate		Univariate	
Characters	HRs (95% CI)	*p*	HRs (95% CI)	*p*	HRs (95% CI)	*p*
Time of HIV diagnosis to breast cancer diagnosis(>12 vs. ≤12 months)	0.37 (0.09–1.55)	0.17	-	-	0.15 (0.02–1.31)	0.09
Neutrophil-to-lymphocyte ratio (High vs. Low group)	5.44 (0.65–45.72)	0.12	-	-	33.31 (0.01–264,621)	0.44
Lymphocyte-to-monocyte ratio (High vs. Low group)	0.22 (0.05–0.93)	0.04	0.15 (0.01–1.65)	0.12	0.20 (0.03–1.28)	0.09
Platelet-to-lymphocyte ratio (High vs. Low group)	42.85 (0.10–17989)	0.22	-	-	44.09 (0.02–90,625.90)	0.33
CD4 (High vs. Low group)	0.31 (0.06–1.58)	0.16	-	-	2.33 (0.38–14.36)	0.36
CD8 (High vs. Low group)	0.01 (0.00–6.62)	0.18	-	-	0.02 (0–27.82)	0.28
CD4/CD8 (High vs. Low group)	35.97 (0.06–20264)	0.27	-	-	44.85 (0.03–81381)	0.32
CD3 (High vs. Low group)	0.10 (0.01–0.90)	0.04	0.35 (0.07–1.74)	0.20	0.15 (0.01–1.6)	0.12

## Data Availability

The datasets used and/or analyzed during the current study are available from the corresponding author on reasonable request.

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
