# Peer review of "Preoperative Biomarkers and Survival in Chinese Breast Cancer Patients with HIV: A Propensity-Score-Matched-Cohort Study"

_viruses, 2023, doi:10.3390/v15071490_

Round 1
Reviewer 1 Report
Comments and Suggestions for Authors
As the life expectancy of HIV-infected patients is improving, morbidity from other diseases is increasing. There is a need to investigate the underlying mechanisms for breast cancer survival disparities by HIV status. Studies on the Chinese population are needed. Wu et al used a propensity score matched sample to assess survival in HIV-infected breast cancer patients from China.
The breast cancer sample consisted of 21 HIV-positive and 396 HIV-negative patients to construct their study cohort of 21 HIV-positive and 21 HIV-negative breast cancer cases. Wu et al results concurred with previous studies showing that HIV-infected breast cancer was more aggressive and had a worse prognosis in their small cohort.
Major problems:
1) Small sample size with multiple comparisons performed.
This is an exploratory study, and the results need to be carefully interpreted. An increase in sample size is needed to be able to better answer your research question.
2) Results presentation
Needs rewrite to improve readability and to clearly present the obtained results.
3) Discussion:
This section will benefit from an extensive rewrite. The small sample size generates a wide 95% CI. The authors need to be careful when interpreting the data. The exploratory nature of the paper needs to be clearly emphasized.
Minor:
1) Better description of the study population at the beginning of the results section is needed with improve data presentation
2) Two significant figures are sufficient
3) There is no need for the P-value when 95% CI is reported
4) The tables need to be redone to improve readability, e.g., clearly separate the cohort into pre-PSM and post-PSM
5) Table 2:
a) Correct to Clinical characteristics.
b)Why are you only showing the Blood biomarkers of HIV+ cases?
6) Line 151: Rewrite this sentence for clarity
7) Line 184: HR and 95% CI needed, not just p-value
8) Line 227: Younger age has been previously been shown to be associated with a more aggressive disease at diagnosis. (Collins LC , Marotti JD , Gelber S , et al : Pathologic features and molecular phenotype by patient age in a large cohort of young women with breast cancer . Breast Cancer Res Treat 131 : 1061 - 1066 , 2012)
9) Line 235: What were your expectations?
Comments on the Quality of English Language
The whole text will benefit from moderate editing of the English language to improve clarity and readability.
Reviewer 2 Report
Comments and Suggestions for Authors
1.The definition of overall survival (OS) in this study, as "death from any cause," may not be appropriate. It would be more suitable to consider breast cancer-related deaths for statistical analysis. Furthermore, the conclusion that "the prognosis for breast cancer in PLWHA was undoubtedly worse" is not adequately supported and needs further justification.
2.The relationship between breast cancer and immunosuppression among PLWHA should be thoroughly explored in the study. The impact of HIV infection on breast cancer incidence, progression, and treatment outcomes needs to be addressed.
3.Despite performing 1:1 Propensity Score Matching, there still appears to be a difference in tumor size between the two groups. This discrepancy requires explanation and consideration, as it may introduce bias into the results.
4.The time interval from HIV diagnosis to breast cancer diagnosis is mentioned in the study. However, it is unclear whether this interval has any correlation with the prognosis of breast cancer.
Additionally, the statement that "PLWHA were diagnosed with breast cancer at a younger age than their HIV-negative counterparts" raises questions about the relationship between HIV and breast cancer incidence.
5. Are there any cases where breast cancer was diagnosed before HIV? The manuscript should provide further discussion on the implications of the temporal relationship between HIV and breast cancer occurrence. Specifically, it would be valuable to address the potential effects of this relationship on breast cancer risk, diagnosis, treatment decisions, prognosis, and psychosocial aspects.
6.The sample size in each group is small, with only 21 patients per group. This limited sample size could introduce bias and affect the reliability of the results. A larger sample size would enhance the statistical power and provide more robust conclusions.
7.It is important to investigate the impact of HIV treatment on the prognosis of breast cancer. Conversely, it would also be valuable to examine how breast cancer treatments such as endocrine therapy, chemotherapy, and targeted therapy affect HIV patients. The interaction and potential consequences of these treatments need to be explored further.
Comments on the Quality of English LanguageThe manuscript would benefit from improving the quality of English writing, as some conclusions are difficult to understand.
Round 2
Reviewer 1 Report
Comments and Suggestions for Authors
The authors have satisfactorily addressed my original comments.
Minor comments:
Table 1: "Femal" remove, Just add all patients were females in the method section.
Comments on the Quality of English LanguageMinor edit needed
Author Response
Thank you for your suggestion, we have revised it accordingly and added it to the method (line 94)
Reviewer 2 Report
Comments and Suggestions for Authors
The authors have provided satisfactory answers to all the questions. However, the small sample size remains a significant concern. The limited number of participants might introduce potential bias and affect the generalizability of the findings. It is important to acknowledge this limitation and consider the impact it may have on the robustness and reliability of the study's conclusions.
Comments on the Quality of English LanguageEnglish language fine.
Author Response
Thank you for your advice on this manuscript, we are fully aware of the limitations of the small sample size and have repeatedly emphasized in our conclusions (line 350)